**Data Availability Statement:** Primary data from the UK Biobank resource are accessible upon application (https://www.ukbiobank.ac.uk/).

# Smoking, alcohol consumption, and cancer: A mendelian randomisation study in UK Biobank and international genetic consortia participants

**Susanna C. Larsson**[1,2]*, **Paul Carter**[3], **Siddhartha Kar**[4], **Mathew Vithayathil**[5], **Amy M. Mason**[6,7], **Karl Michaëlsson**[1], **Stephen Burgess**[3,8]

**1** Department of Surgical Sciences, Uppsala University, Uppsala, Sweden, **2** Unit of Cardiovascular and Nutritional Epidemiology, Institute of Environmental Medicine, Karolinska Institutet, Stockholm, Sweden, **3** Department of Public Health and Primary Care, University of Cambridge, Cambridge, United Kingdom, **4** MRC Integrative Epidemiology Unit, Bristol Medical School, University of Bristol, Bristol, United Kingdom, **5** MRC Cancer Unit, University of Cambridge, Cambridge, United Kingdom, **6** British Heart Foundation Cardiovascular Epidemiology Unit, Department of Public Health and Primary Care, University of Cambridge, Cambridge, United Kingdom, **7** National Institute for Health Research Cambridge Biomedical Research Centre, University of Cambridge and Cambridge University Hospitals, Cambridge, United Kingdom, **8** MRC Biostatistics Unit, University of Cambridge, Cambridge, United Kingdom

* susanna.larsson@ki.se

## Abstract

### Background

Smoking is a well-established cause of lung cancer and there is strong evidence that smoking also increases the risk of several other cancers. Alcohol consumption has been inconsistently associated with cancer risk in observational studies. This mendelian randomisation (MR) study sought to investigate associations in support of a causal relationship between smoking and alcohol consumption and 19 site-specific cancers.

### Methods and findings

We used summary-level data for genetic variants associated with smoking initiation (ever smoked regularly) and alcohol consumption, and the corresponding associations with lung, breast, ovarian, and prostate cancer from genome-wide association studies consortia, including participants of European ancestry. We additionally estimated genetic associations with 19 site-specific cancers among 367,643 individuals of European descent in UK Biobank who were 37 to 73 years of age when recruited from 2006 to 2010. Associations were considered statistically significant at a Bonferroni corrected $p$-value below 0.0013. Genetic predisposition to smoking initiation was associated with statistically significant higher odds of lung cancer in the International Lung Cancer Consortium (odds ratio [OR] 1.80; 95% confidence interval [CI] 1.59–2.03; $p = 2.26 \times 10^{-21}$) and UK Biobank (OR 2.26; 95% CI 1.92–2.65; $p = 1.17 \times 10^{-22}$). Additionally, genetic predisposition to smoking was associated with statistically significant higher odds of cancer of the oesophagus (OR 1.83; 95% CI 1.34–2.49; $p = 1.31 \times 10^{-4}$), cervix (OR 1.55; 95% CI 1.27–1.88; $p = 1.24 \times 10^{-5}$), and bladder

Derived data supporting the results of this study are available in the S1 Data file.

**Funding:** SCL reports support from the Swedish Research Council for Health, Working Life and Welfare (2018-00123), the Swedish Research Council (2019-00977), and the Swedish Heart-Lung Foundation (20190247). SK reports support from a Cancer Research UK programme grant, the Integrative Cancer Epidemiology Programme (C18281/A19169) and a Junior Research Fellowship from Homerton College, Cambridge. AMM is funded by the National Institute for Health Research [Cambridge Biomedical Research Centre at the Cambridge University Hospitals NHS Foundation Trust]. SB reports support from a Sir Henry Dale Fellowship jointly funded by the Wellcome Trust and the Royal Society (204623/Z/16/Z). The funders had no role in study design, data collection and analysis, decision to publish, or preparation of the manuscript.

**Competing interests:** I have read the journal's policy and the authors of this manuscript have the following competing interests: SB is a paid statistical consultant on *PLOS Medicine*'s statistical board.

**Abbreviations:** CI, confidence interval; MR, mendelian randomisation; MR-PRESSO, mendelian randomisation pleiotropy residual sum and outlier; OR, odds ratio; SNP, single-nucleotide polymorphism; STROBE, Strengthening the Reporting of Observational Studies in Epidemiology.

(OR 1.40; 95% CI 1.92–2.65; $p = 9.40 \times 10^{-5}$) and with statistically nonsignificant higher odds of head and neck (OR 1.40; 95% CI 1.13–1.74; $p = 0.002$) and stomach cancer (OR 1.46; 95% CI 1.05–2.03; $p = 0.024$). In contrast, there was an inverse association between genetic predisposition to smoking and prostate cancer in the Prostate Cancer Association Group to Investigate Cancer Associated Alterations in the Genome consortium (OR 0.90; 95% CI 0.83–0.98; $p = 0.011$) and in UK Biobank (OR 0.90; 95% CI 0.80–1.02; $p = 0.104$), but the associations did not reach statistical significance. We found no statistically significant association between genetically predicted alcohol consumption and overall cancer ($n = 75,037$ cases; OR 0.95; 95% CI 0.84–1.07; $p = 0.376$). Genetically predicted alcohol consumption was statistically significantly associated with lung cancer in the International Lung Cancer Consortium (OR 1.94; 95% CI 1.41–2.68; $p = 4.68 \times 10^{-5}$) but not in UK Biobank (OR 1.12; 95% CI 0.65–1.93; $p = 0.686$). There was no statistically significant association between alcohol consumption and any other site-specific cancer. The main limitation of this study is that precision was low in some analyses, particularly for analyses of alcohol consumption and site-specific cancers.

## Conclusions

Our findings support the well-established relationship between smoking and lung cancer and suggest that smoking may also be a risk factor for cancer of the head and neck, oesophagus, stomach, cervix, and bladder. We found no evidence supporting a relationship between alcohol consumption and overall or site-specific cancer risk.

## Author summary

### Why was this study done?

- Tobacco smoking and alcoholic beverage consumption are common addictive behaviours and important risk factors for mortality.

- Observational evidence has shown that smoking is a risk factor for cancers of the lung, bladder, kidney, gastrointestinal tract, and cervix, but uncertainty persists about the causal role of smoking for the development of other cancers.

- The causal role of alcohol consumption for site-specific cancers is uncertain, as available evidence originates from observational studies which are susceptible to confounding and reverse causation bias.

### What did the researchers do and find?

- Using the mendelian randomisation (MR) design, we found that genetic predisposition to smoking is associated with a statistically significant increased risk of cancer of the lung, oesophagus, cervix, and bladder and with a statistically nonsignificant increased risk of head and neck and stomach cancer at the Bonferroni-corrected significance threshold.

- Genetically predicted alcohol consumption was statistically significantly positively associated with lung cancer but not with any other site-specific cancer or overall cancer.

### What do these findings mean?

- In this study, we observed a relationship between smoking and lung cancer, as well as evidence that smoking may also be a risk factor for cancer of the head and neck, oesophagus, stomach, cervix, and bladder.
- We found no evidence supporting a relationship between alcohol consumption and overall or site-specific cancer risk.

## Introduction

Tobacco smoking and alcoholic beverage consumption are common addictive behaviours and important causes of mortality [1–3]. The causal link between smoking and risk of lung cancer is well established [4,5]. Strong experimental and observational evidence also indicates that smoking is a risk factor for cancers of the bladder, kidney, gastrointestinal tract (head and neck, oesophagus, stomach, colorectum, pancreas, and liver) and cervix [4–8], but uncertainty persists about the causal role of smoking for the development of other cancers.

Observational data on alcohol consumption in relation to various cancers are contrasting [7–14]. Alcohol consumption has been reported to be positively associated with risk of cancers of the head and neck, oesophagus, stomach, liver, and breast [7,9,12–14] but inversely associated with kidney cancer [9,11] and non-Hodgkin lymphoma [9,12]. With regard to colorectal cancer, a recent meta-analysis of 16 cohort studies showed that heavy alcohol consumption was associated an increased risk of colorectal cancer, whereas light-to-moderate drinking was associated with a decreased risk [10]. Given that much of the current evidence originates from observational epidemiological studies, the causal nature of these findings needs assessment. This is particularly important in the context of these addictive health behaviours, which can be open to confounding by factors such as other destructive health behaviours and socioeconomic status.

A recent meta-analysis of genome-wide association studies identified a number of single-nucleotide polymorphisms (SNPs) associated with smoking and alcohol consumption [15]. Those SNPs can be used as instruments for these exposures in mendelian randomisation (MR) analyses to infer causality. The MR technique is based on Mendel's law of independent assortment, which states that the inheritance of one characteristic is independent of the inheritance of another. Thus, levels of the exposure predicted by the SNPs are usually independent of other exposures, thereby reducing confounding in MR studies. In addition, the MR study design avoids reverse causality because disease development cannot affect genotype.

The primary aim of this study was to use MR to examine the associations of smoking and alcohol consumption with 19 site-specific cancers using data from four large-scale genome-wide association studies consortia and UK Biobank. In complementary analyses, we assessed the associations of genetically predicted smoking and alcohol consumption with overall cancer in UK Biobank to determine the overall impact of smoking and alcohol from a public health perspective.

## Methods

### Outcome data sources

Publicly available summary-level data for lung, breast, ovarian, and prostate cancer were obtained respectively from the International Lung Cancer Consortium (11,348 cases and 15,861 controls) [16], the Breast Cancer Association Consortium (122,977 cases and 105,974 controls) [17], the Ovarian Cancer Association Consortium (25,509 cases and 40,941 controls) [18], and the Prostate Cancer Association Group to Investigate Cancer Associated Alterations in the Genome consortium (79,148 cases and 61,106 controls) [19]. All participants included in the consortia were of European ancestry and came from European and North American countries and Australia. The lung cancer consortium included both women and men, whereas the breast and ovarian cancer consortia included women only and the prostate cancer consortium men only. Data from the consortia were extracted through the MR-Base platform [20].

In addition, we estimated genetic associations with 19 site-specific cancers (with at least 400 cases) and overall cancer in UK Biobank, a cohort study of about 500,000 adults (37 to 73 years of age at baseline) recruited between 2006 and 2010 [21]. In the current analyses, we included 367,643 UK Biobank participants of European descent after exclusion of participants with other ethnicities (to reduce population stratification bias), those with relatedness of third degree or higher, excess heterozygosity, and low genotype call rate. Cancer cases were ascertained until March 31, 2017, and were defined based on data from national registries (International Classification of Diseases, 9th and 10th revision codes) and self-reported information verified by interview with a nurse (S1 Table). We calculated beta coefficients and standard errors of the genetic associations with cancer using logistic regression, with adjustment for age, sex, and 10 genetic principal components. The UK Biobank study was approved by the North West Multicenter Research Ethics Committee. All participants provided written informed consent. The present analyses were approved by the Swedish Ethical Review Authority.

### Genetic instruments

Instrumental variables for the exposures were selected from a meta-analysis of genome-wide association studies with a total of 1,232,091 individuals of European descent in the analysis of smoking initiation (i.e., probability of ever smoked regularly) and 941,280 individuals of European descent in the analysis of alcohol consumption [15]. A total of 378 and 99 conditionally independent SNPs associated with smoking initiation and alcohol consumption (log-transformed alcoholic drinks per week), respectively, at the genome-wide significance threshold ($p < 5 \times 10^{-8}$) were identified [15]. Linkage disequilibrium (defined as $R^2 > 0.1$) between SNPs was assessed using LDlink [22] and was detected among 16 SNP pairs for smoking initiation and among four SNP pairs for alcohol consumption. The SNP with the weakest association with the exposure was removed. SNPs that were unavailable in the outcome datasets were replaced by a suitable proxy (minimum linkage disequilibrium $R^2 = 0.8$) where available. Our genetic instruments comprised between 346 and 361 independent SNPs for smoking initiation and between 89 and 94 independent SNPs for alcohol consumption. The SNPs explained 2.3% and 0.2%–0.3% of the variation in smoking initiation and alcohol consumption, respectively [15]. The genetic instruments were strongly associated with the exposures in UK Biobank, with an F statistic from regression of the exposure on the variants of about 75 for smoking initiation and 19–29 for alcohol consumption. The two exposures were moderately genetically correlated ($r_g = 0.36$) [15]. The effect sizes are expressed per standard deviation increase in the exposure [15]. For smoking, this was calculated from the weighted average prevalence of ever smokers across the studies included in the meta-analysis [15].

## Statistical analysis

The principal analyses were conducted using the inverse-variance weighted approach (under a multiplicative random-effects model), which provides the most precise estimates but assumes that all SNPs are valid instrumental variables [23]. In sensitivity analyses, the following approaches were applied: (1) multivariable MR analysis (inverse-variance weighted method) with smoking adjusted for alcohol consumption and vice versa; (2) weighted median method, which provides a causal estimate if at least 50% of the weight in the analysis comes from valid instrumental variables [23]; (3) contamination mixture method, which performs MR robustly and efficiently in the presence of invalid instrumental variables [24]; (4) MR pleiotropy residual sum and outlier (MR-PRESSO) method, which can detect and adjust for horizontal pleiotropy by outlier removal [25]; and (5) MR-Egger regression method, which can detect and adjust for directional pleiotropy but has low precision [23]. The mrrobust [26], Mendelian-Randomization [27], and MRPRESSO [25] packages were used for the statistical analyses. The reported odds ratios (OR) are per one standard deviation increase in the prevalence of smoking initiation and per standard deviation increase in log-transformed alcoholic drinks per week. We calculated the power at different ORs for each cancer site [28]. All statistical tests were 2-tailed. Associations were considered statistically significant at a Bonferroni corrected $p$-value below 0.0013 (correcting for 2 exposures and 19 outcomes).

## Protocol

There was no formal predefined protocol or prospective analysis plan for this study. The MR analyses of genetic predisposition to smoking and alcohol consumption in relation to overall and site-specific cancers in UK Biobank were initiated in August 2019. Following peer review, we (i) conducted MR analyses of smoking and alcohol consumption in relation to lung, breast, ovarian, and prostate cancer using publicly available data from international consortia for these cancers; (ii) defined the analysis of site-specific cancer as the primary analysis and applied Bonferroni correction to site-specific cancers; (iii) calculated statistical power for each site-specific cancer; and (iv) performed a complementary analysis using the contamination mixture method.

This study is reported as per the Strengthening the Reporting of Observational Studies in Epidemiology (STROBE) guideline (S1 Checklist).

# Results

In analyses using consortia data, we had 80% power at a significance level of 0.05 to detect an OR ranging from 1.05 (for breast cancer) to 1.25 (for lung cancer) for smoking initiation, and a corresponding OR ranging from 1.28 to 1.90 for alcohol consumption (S2 Table). Relatively strong magnitude of associations was necessary to detect significant ORs in site-specific cancer analyses based on UK Biobank data, particularly for alcohol consumption (S2 Table).

Genetic predisposition to smoking initiation was associated with a statistically significant higher odds of lung cancer in the International Lung Cancer Consortium (OR 1.80; 95% confidence interval [CI] 1.59–2.03; $p = 2.26 \times 10^{-21}$) and UK Biobank (OR 2.26; 95% CI 1.92–2.65; $p = 1.17 \times 10^{-22}$) (Fig 1). Additionally, genetic predisposition to smoking was associated with statistically significantly higher odds of cancer of the oesophagus (OR 1.83; 95% CI 1.34–2.49; $p = 1.31 \times 10^{-4}$), cervix (OR 1.55; 95% CI 1.27–1.88; $p = 1.24 \times 10^{-5}$), and bladder (OR 1.40; 95% CI 1.92–2.65; $p = 9.40 \times 10^{-5}$) and with statistically nonsignificant higher odds of head and neck (OR 1.40; 95% CI 1.13–1.74; $p = 0.002$) and stomach cancer (OR 1.46; 95% CI 1.05–2.03; $p = 0.024$) (Fig 1). In contrast, there was a statistically nonsignificant inverse association between genetic predisposition to smoking and prostate cancer in the Prostate Cancer

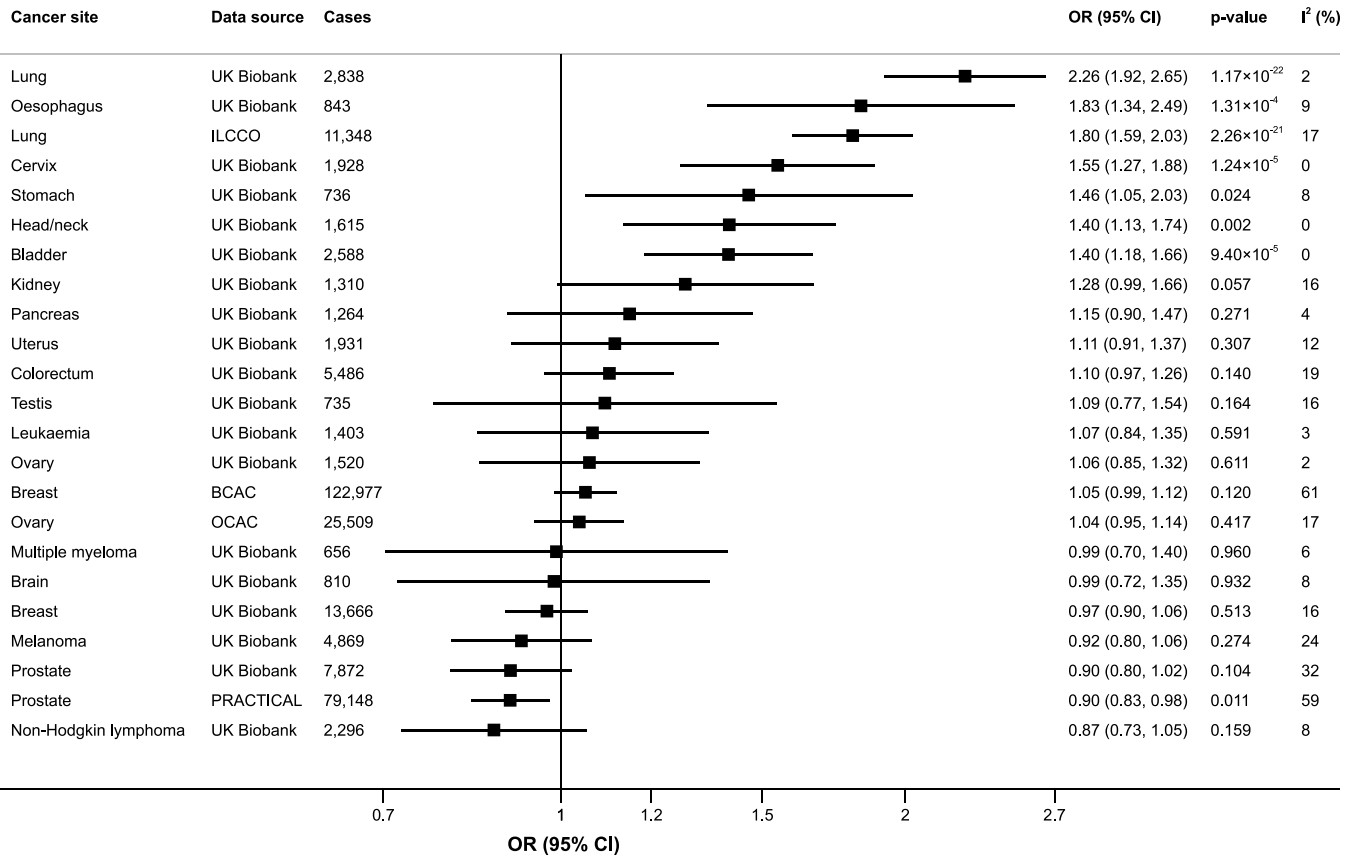

**Fig 1. Associations of genetic predisposition to smoking initiation with site-specific cancers.** ORs are per one standard deviation increase in probability of smoking initiation (ever smoked regularly). Results are obtained from the random-effects inverse-variance weighted method. The $I^2$ statistic quantifies the amount of heterogeneity among estimates based on individual SNPs. BCAC, Breast Cancer Association Consortium; CI, confidence interval; ILCCO, International Lung Cancer Consortium; OCAC, Ovarian Cancer Association Consortium; OR, odds ratio; PRACTICAL, Prostate Cancer Association Group to Investigate Cancer Associated Alterations in the Genome consortium; SNP, single-nucleotide polymorphism.

Association Group to Investigate Cancer Associated Alterations in the Genome consortium (OR 0.90; 95% CI 0.83–0.98; $p$ = 0.011) and in UK Biobank (OR 0.90; 95% CI 0.80–1.02; $p$ = 0.104) (Fig 1). Results were consistent in sensitivity analyses (S3 Table). Genetic predisposition to smoking initiation was associated with a statistically significant higher odds of both lung adenocarcinoma and squamous cell lung cancer but was not associated with significantly higher odds of oestrogen receptor positive or negative breast tumours or with any subtype of non-Hodgkin lymphoma or leukaemia (S4 Table).

Genetically predicted alcohol consumption was statistically significantly positively associated with lung cancer in the International Lung Cancer Consortium (OR 1.94; 95% CI 1.41–2.68; $p$ = 4.68 × 10$^{-5}$) but not in UK Biobank (OR 1.12; 95% CI 0.65–1.93; $p$ = 0.686) (Fig 2). After adjustment for genetic predisposition to smoking using multivariable MR, the association between genetically predicted alcohol consumption and lung cancer in the International Lung Cancer Consortium was attenuated and nonsignificant (OR 1.75; 95% CI 1.23–2.49; $p$ = 0.002) (S5 Table). There was no statistically significant association between genetically predicted alcohol consumption and any other site-specific cancer (Fig 2). However, the precision

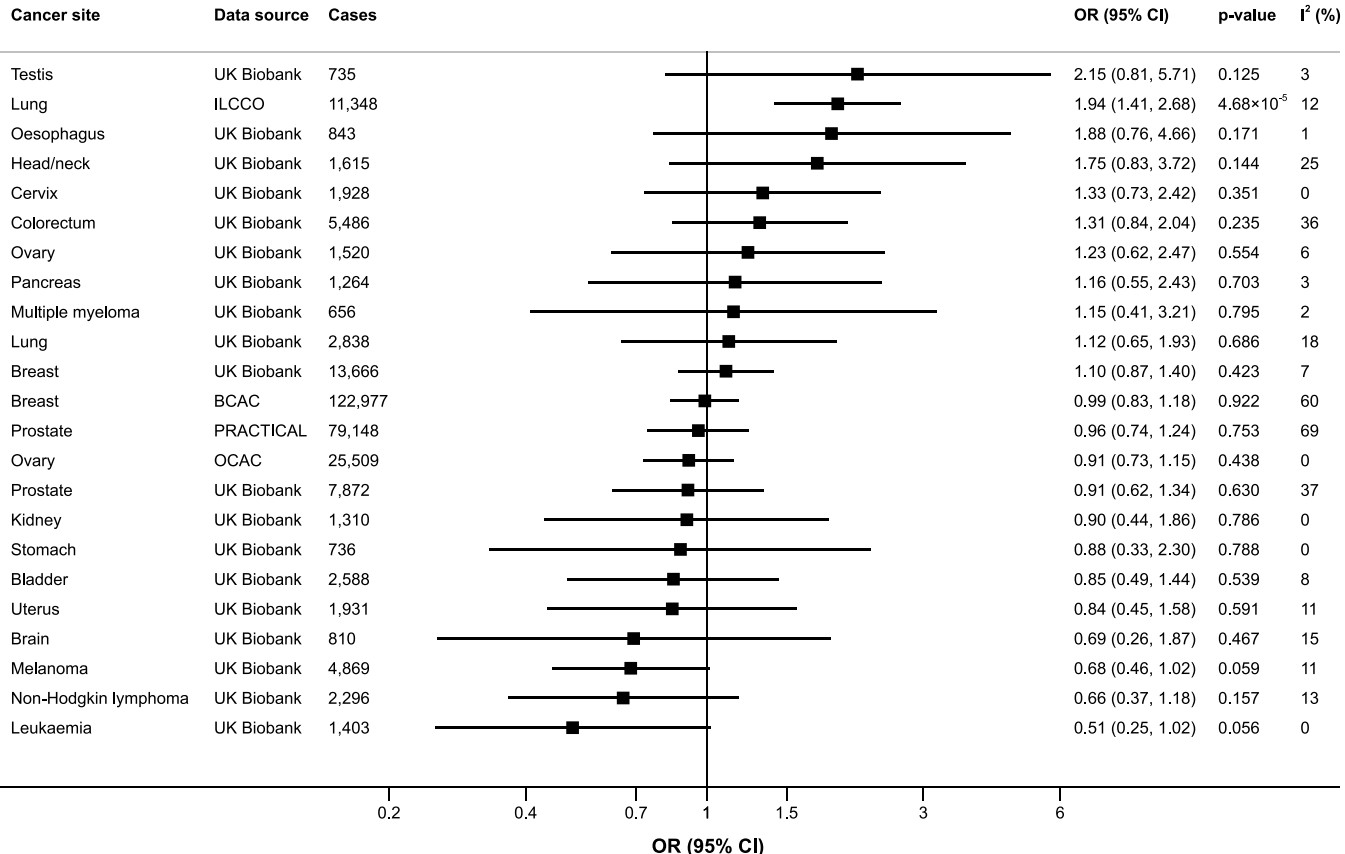

**Fig 2. Associations of genetically predicted alcohol consumption with site-specific cancers.** ORs are per one standard deviation increase of log-transformed alcoholic drinks per week. Results are obtained from the multiplicative random-effects inverse-variance weighted method. The $I^2$ statistic quantifies the amount of heterogeneity among estimates based on individual SNPs. BCAC, Breast Cancer Association Consortium; CI, confidence interval; ILCCO, International Lung Cancer Consortium; OCAC, Ovarian Cancer Association Consortium; OR, odds ratio; PRACTICAL, Prostate Cancer Association Group to Investigate Cancer Associated Alterations in the Genome consortium; SNP, single-nucleotide polymorphism.

of the estimates was generally low and the ORs were above 1.50 for testicular, head and neck, and oesophageal cancers and below 0.70 for leukaemia, non-Hodgkin lymphoma, melanoma, and brain cancer (Fig 2). The ORs were consistently above 2 for testicular cancer in all sensitivity analyses, but none of the associations reached statistical significance (S5 Table). Genetically predicted alcohol consumption was associated with statistically nonsignificant higher odds of both lung adenocarcinoma and squamous cell lung cancer in the International Lung Cancer Consortium but was not associated with any lung cancer subtype in UK Biobank, oestrogen receptor positive or negative breast tumours, or any subtype of non-Hodgkin lymphoma or leukaemia (S6 Table).

For overall cancer ($n$ = 75,037 cases, also including cancers with <400 cases, in UK Biobank) a statistically nonsignificant positive association was observed with genetic predisposition to smoking in the inverse-variance weighted analysis (OR 1.06; 95% CI 1.02–1.10; $p$ = 0.007), but the association was attenuated in the alcohol-adjusted analysis (OR 1.03; 95% CI 0.98–1.08; $p$ = 0.304) (S3 Table). Genetically predicted alcohol consumption was not associated with overall cancer (OR 0.95; 95% CI 0.84–1.07; $p$ = 0.376) (S5 Table).

## Discussion

In the present MR study, we systematically assessed the associations of genetically predicted smoking and alcohol consumption with a broad range of cancer types. Our results showed that genetic predisposition to smoking was associated with an increased risk of several cancers, supporting a causal relationship as consistent with previous observational studies. Genetically predicted alcohol consumption was positively associated with lung cancer in the International Lung Cancer Consortium but was not statistically significantly associated with any other site-specific cancer or overall cancer. However, the precision was low in the site-specific cancer analyses and there were relatively strong but statistically nonsignificant positive associations of genetically predicted alcohol consumption (ORs above 1.5) with testicular, head and neck, and oesophageal cancer, as well as inverse associations (ORs below 0.7) with leukaemia, non-Hodgkin lymphoma, melanoma, and brain cancer.

Both tobacco and alcohol are biologically plausible tumour-promoting behaviours. Tobacco smoke can increase the risk of cancer through its content of carcinogens, such as nitrosamines, polycyclic aromatic hydrocarbons, acrylamines, volatile organics, and cadmium [7]. The carcinogenic effect of smoking on lung cancer is well established, and our MR findings corroborate the observational evidence that smoking is also a risk factor for cancers of the head and neck, oesophagus, stomach, cervix, and bladder [4–7]. Smoking is the major risk factor for bladder cancer, and it has been estimated that ever smoking accounts for about two thirds and one third of all bladder cancer cases in men and women, respectively [6]. Conventional observational studies have further shown that smoking increases the risk of kidney, pancreatic, and colorectal cancer [4,5,7]. Although we failed to detect significant associations of genetic predisposition to smoking with those cancers, the associations were positive, with an OR of 1.28 for kidney cancer, 1.15 for pancreatic cancer, and 1.10 for colorectal cancer. Similarly, smoking has been found to have a protective effect on melanoma risk in multiple meta-analyses, and although our estimate was in this direction (OR 0.92), the association did not reach statistical significance. Conventional observational studies have reported inconclusive results or no association of smoking with breast, ovarian, and brain cancers, melanoma, non-Hodgkin lymphoma, leukaemia, and multiple myeloma [4,5,7,29,30]. This MR study provided no evidence of a causal association between smoking and those cancers. Furthermore, we found no support of an inverse association between smoking and uterine cancer, consistent with findings from most observational studies [5,7]. However, we found a statistically nonsignificant inverse association between smoking and prostate cancer. It is worth interpreting this result in light of the fact that only 20% of the prostate cancer cases included in the prostate cancer consortium were of advanced or highly aggressive disease [19]. Genetic association estimates specific to advanced or highly aggressive prostate cancer were not publicly available. Our MR results for prostate cancer, based largely on non-advanced disease, are consistent with the inverse association between smoking and prostate cancer risk that has been reported specifically for localised and low-grade disease in observational studies [31,32]. The inverse association may reflect detection bias, such that smokers may be less likely to undergo prostate-specific antigen screening and therefore are not diagnosed with prostate cancer until a late stage or not at all.

Alcohol consumption may increase the risk of cancer through its oxidised metabolite acetaldehyde, which is carcinogenic to humans [7]. Alcohol consumption might also lower cancer risk by other mechanisms, such as increased insulin sensitivity through increased adiponectin levels [33]. Red wine in particular has been suggested as potentially anticarcinogenic due to its flavonoid content, which is both anti-inflammatory and antioxidative [34,35]. The present MR findings are consistent with those of observational studies [6,7,9,12–14] and previous MR studies [36,37] indicating that alcohol consumption increases the risk of cancers of the head and

neck and oesophagus, although our estimates had low precision and did not reach statistical significance. Alcohol drinking has also been reported to be associated with breast cancer risk in a dose-response manner, with 8% to 12% increase in risk per 10 g/day increase of alcohol consumption [9,13,14]. This MR study was underpowered to detect such a relatively modest association. A harmful effect of heavy drinking on colorectal cancer has been suggested by observational studies [10,12] as well as an MR study using the *ALDH2* genotype as a marker of alcohol exposure [38]. In contrast, light alcohol consumption has been shown to be unrelated [12,14] or inversely [10] associated with colorectal cancer risk. If anything, our findings indicated a positive association of alcohol consumption with colorectal cancer risk, but we were unable to investigate a potential nonlinear relation. This study could not confirm an inverse association of alcohol consumption with colorectal [10] and kidney [9,11] cancer and non-Hodgkin lymphoma [9,12], but the association with non-Hodgkin lymphoma was in the same direction. Our results agree with observational studies, which have shown no consistent relation between alcohol consumption and risk of leukaemia, melanoma, and brain cancer [9,12,39], and with a previous MR study demonstrating no association between alcohol exposure and prostate cancer incidence [40]. Likewise, data on alcohol consumption in relation to testicular cancer risk are limited. The strong though nonsignificant positive association between genetically predicted alcohol consumption and testicular cancer needs confirmation by larger MR studies.

The inconsistent results for alcohol consumption and lung cancer between the International Lung Cancer Consortium and UK Biobank may simply be a chance finding. Another possibility is that the association of genetically predicted alcohol consumption with lung cancer may be due to smoking rather than alcohol. While we were able to adjust for individual smoking behaviour through multivariable MR analysis, increased alcohol consumption may lead to greater exposure to cigarette smoke via passive smoking. This mechanism would be less represented in UK Biobank due to the relatively low smoking rate in UK Biobank, and the smoking ban in the UK.

Key strengths of this study include the MR design, which mitigated bias due to confounding and reverse causality, and the use of multiple instrumental variables for smoking and alcohol consumption, which allowed sensitivity analyses to identify and adjust for pleiotropy. Another strength is that the associations of smoking and alcohol consumption with cancer at many sites could be assessed in a single large population of individuals of European descent. This enabled comparison of the magnitude of the adverse impact of smoking and alcohol consumption on different cancers while minimising population stratification bias.

A major limitation of this study is that the precision was low in some analyses. Analyses of alcohol consumption in particular had low precision owing to low variance explained by the SNPs, albeit with an F statistic above the conventional cutoff of 10. A low power may explain the lack of statistically significant association of genetically predicted smoking and alcohol consumption with certain site-specific cancers. For overall analyses, while overall cancer is a combination of different malignancies that may have different underlying aetiologies, all cancers are known to share common underlying "hallmark" molecular and cellular aberrations. Combining these malignancies together means that our endpoint is dependent on the characteristics of the analytic sample and the relative prevalence of different cancer types. In particular, cancers with greater survival chances will be overrepresented in the case sample. However, for most individuals, the choice to smoke cigarettes or drink alcohol is likely to be guided by the impact on the overall risk of cancer, not any particular site-specific cancer. Hence, we believe the results for overall cancer have direct relevance for public health. Another shortcoming is that we were unable to assess possible U- or J-shaped relations between alcohol consumption and cancer, or differential effects of specific alcoholic beverage types or drinking patterns. A further potential limitation is that UK Biobank participants were included in both

the exposure and outcome datasets, which might have introduced some bias in the MR estimates in the direction of the estimates of observational studies. Nonetheless, only around one third of participants in the genome-wide association studies of smoking and alcohol consumption came from the UK Biobank study and the genetic instruments were relatively strongly related to the exposures (F statistic >10), implying that bias from participant overlap is relatively small [41]. An additional limitation is that smoking initiation is a binary exposure. An MR estimate with a binary exposure and binary outcome is difficult to interpret as a specific causal effect [42]. Furthermore, the MR estimate cannot simply be compared with the association between self-reported smoking and cancer risk estimated by observational studies. However, even in this setting, MR estimates are unbiased under the null [43] and thus provide a valid test of the causal null hypothesis even if the estimates do not reflect meaningful causal parameters. Here, we expressed all estimates in terms of the association between genetically predicted levels of smoking initiation and the outcome, rather than making claims about the numerical magnitude of the causal effect. Our analyses included individuals of European descent and therefore might not be generalisable to other populations.

These analyses broaden the evidence base for the harmful effect of cigarette smoking, which has previously been demonstrated for most cardiovascular diseases [44,45], type 2 diabetes [46], and bone fracture [47]. Those findings along with results of the present study provide further evidential support for public health interventions to reduce cigarette smoking initiation in the population, and suggest that these strategies will have an important impact on lessening the burden of major diseases, and of cancer in particular.

## Conclusions

The results of this study support the well-established relationship between smoking and lung cancer, and suggest that smoking may also be a risk factor for cancer of the head and neck, oesophagus, stomach, cervix, and bladder. We found no evidence in support of a relationship between alcohol consumption and overall cancer risk, but associations between alcohol consumption and risk of site-specific cancer should be further investigated.

## Supporting information

**S1 STROBE Checklist. STROBE, Strengthening the Reporting of Observational Studies in Epidemiology.**
(DOCX)

**S1 Table. Sources and definition of cancers in UK Biobank.**
(XLSX)

**S2 Table. Power calculation for the associations of smoking and alcohol consumption with cancer.**
(XLSX)

**S3 Table. Associations of genetic predisposition to smoking initiation with site-specific cancers in the primary inverse-variance weighted analysis and in sensitivity analyses using other MR methods.** MR, mendelian randomisation.
(XLSX)

**S4 Table. Associations of genetic predisposition to smoking initiation with subtypes of lung cancer, non-Hodgkin lymphoma, and leukaemia.**
(XLSX)

**S5 Table. Associations of genetically predicted alcohol consumption with site-specific cancers in the primary inverse-variance weighted analysis and in sensitivity analyses using other MR methods.** MR, mendelian randomisation.
(XLSX)

**S6 Table. Associations of genetically predicted alcohol consumption with subtypes of lung cancer, breast cancer, non-Hodgkin lymphoma, and leukaemia in MR analyses.** MR, mendelian randomisation.
(XLSX)

**S1 Data. Derived summary statistics data supporting the result of this study.**
(XLSX)

## Acknowledgments

This research has been conducted using the UK Biobank Resource under Application number 29202.

   **Disclaimer:** The views expressed are those of the authors and not necessarily those of the NHS, the NIHR, or the Department of Health and Social Care.

## Author Contributions

**Conceptualization:** Susanna C. Larsson, Paul Carter, Siddhartha Kar, Mathew Vithayathil, Amy M. Mason, Karl Michaëlsson, Stephen Burgess.

**Data curation:** Susanna C. Larsson, Amy M. Mason, Stephen Burgess.

**Formal analysis:** Susanna C. Larsson.

**Funding acquisition:** Susanna C. Larsson, Stephen Burgess.

**Investigation:** Susanna C. Larsson, Paul Carter, Siddhartha Kar, Mathew Vithayathil, Amy M. Mason, Karl Michaëlsson, Stephen Burgess.

**Methodology:** Susanna C. Larsson, Paul Carter, Siddhartha Kar, Mathew Vithayathil, Amy M. Mason, Stephen Burgess.

**Project administration:** Susanna C. Larsson.

**Resources:** Susanna C. Larsson.

**Software:** Susanna C. Larsson.

**Supervision:** Susanna C. Larsson.

**Validation:** Susanna C. Larsson.

**Visualization:** Susanna C. Larsson.

**Writing – original draft:** Susanna C. Larsson.

**Writing – review & editing:** Susanna C. Larsson, Paul Carter, Siddhartha Kar, Mathew Vithayathil, Amy M. Mason, Karl Michaëlsson, Stephen Burgess.

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
