## [Decision Letter · Decision Letter 0]

12 Mar 2020

Dear Dr. Larsson,

Thank you very much for submitting your manuscript "Smoking, alcohol consumption and cancer in UK Biobank: A Mendelian randomisation study" (PMEDICINE-D-19-03758) for consideration at PLOS Medicine. 

[LINK]

In light of these reviews, I am afraid that we will not be able to accept the manuscript for publication in the journal in its current form, but we would like to consider a revised version that addresses the reviewers' and editors' comments. Obviously we cannot make any decision about publication until we have seen the revised manuscript and your response, and we plan to seek re-review by one or more of the reviewers. 

We expect to receive your revised manuscript by Apr 02 2020 11:59PM. Please email us (plosmedicine@plos.org) if you have any questions or concerns.

We look forward to receiving your revised manuscript. 

Sincerely,

Caitlin Moyer, Ph.D.

Associate Editor 

PLOS Medicine

plosmedicine.org

Ref 1 raises important points, which we think are especially important to address. 

General point: please tone down sentences such as this in the abstract: “These findings indicate that smoking is causally associated with an increased risk of cancer”. While MR study is indicative, it cannot show causality and so please change this sentence and other instances in the text (including the Author Summary and the conclusion section). 

Please provide p values in the abstract with 95%Cis. In addition, is it possible to include some summary demographic information in the abstract? 

Please remove spaces in between multiple refs in square brackets.

Please use sections and Paragraphs for the STROBE – pages change during revisions and formatting etc. 

Comments from the reviewers:

Reviewer #1: Larsson et al. use Mendelian randomisation (MR) to examine the relationship between smoking and alcohol consumption, and 19 site-specific cancers. The manuscript is concise and well written, and the methodologies applied are appropriate. The key limitation of the study is its low power to identify causal relationships with many of the cancers. This is especially true for alcohol consumption, which the authors causally relate to no site-specific cancer, despite strong prior evidence from other studies. This low power prevents the study from providing further information on cancers for which evidence of a relationship with smoking and alcohol consumption has been so far mixed. 

Major comments: 

1) Low power to detect causal relationships occurs due to the authors use of UK BioBank data for the site-specific cancers. I think this is the main issue with this study. For many cancers, much larger datasets have been published and are publicly available (for example the BCAC breast cancer GWAS contains >100,000 breast cancer cases, in contrast to the ~14,000 breast cancer cases in UK BioBank). Where possible, these larger cancer GWAS should be used in place of the UK BioBank data. I appreciate that in some instances data access limitations and scientific politics would prevent the authors of this manuscript from accessing these larger data sets. However, use of larger datasets where possible would substantially improve this analysis. Currently, the analysis of alcohol consumption with site-specific cancer risk feels like a missed opportunity, given that for 13/19 site-specific cancer, the study has less than 80% power to detect ORs of 3. 

2) Where UK BioBank data cannot be replaced with data from a larger cancer GWAS, further information should be provided about how cancer association statistics were computed using UK BioBank data. Did the authors compute these statistics themselves, or were they obtained from another source, such as the Neale Lab (who have computed association statistics for UK BioBank data http://www.nealelab.is/uk-biobank). Were any samples excluded for QC reasons? 

3) The authors note that sample overlap between the exposure and outcome datasets is likely biasing their results. Ideally, the authors should recompute association statistics for smoking and alcohol consumption excluded UK BioBank participants. If this is not possible, then this caveat should be more prominent, preferably in the abstract. The potential magnitude of this bias should also be estimated, as per Burgess et al. (https://www.ncbi.nlm.nih.gov/pubmed/27625185). 

4) We have previously observed that wald-type ratio estimators do not provide accurate estimates of the causal OR when both the exposure and outcome traits are binary (https://www.ncbi.nlm.nih.gov/pubmed/29540232, full disclosure: I am an author on this paper). This has also been noted by others (e.g. Palmer et al. https://www.ncbi.nlm.nih.gov/pubmed/21555716). Are the causal estimates between smoking initiation and cancer risk reported here potentially affected by such bias? If so, this should be noted. 

Minor comments: 

1) Supplementary Figure 1 should be annotated with the site-specific cancers, to make it easier for readers to identify the cancers for which the study has suitable power. 

2) The authors consider their analysis of overall cancer risk to be their "primary" analysis, whilst the analysis of site-specific cancers are "secondary" analyses. How smoking affects cancer risk differs substantially between sites (as seen in Figure 2), and I am therefore unsure how useful the analysis of overall cancer risk is. The analysis of site-specific cancers is potentially more interesting, and therefore this should be the "primary" analysis. 

3) The conclusion currently states: "These MR findings indicate smoking is causally associated with an increased risk of cancer, particularly of the lung...". The causal association between smoking and lung cancer is very well understood and more focus should therefore be placed on the site-specific cancers over which there is more debate. 

4) Further information should be provided about how the lifetime smoking IV takes into account duration, heaviness etc. 

5) The authors apply Bonferroni correction to overall cancer risk, but not the site-specific cancers. Multiple testing should be corrected for in both analyses. 

Alex Cornish (ICR, London) - following Stephen Burgess' lead of open peer review.

Reviewer #2: This is a two-sample MR analysis that investigates the association of smoking initiation, lifetime smoking and alcohol consumption with several cancers in the UK Biobank. This is the largest MR study investigating smoking, alcohol in relation to cancer risk. It has verified several positive associations observed in the observational literature for smoking and risk of several cancers, but failed to do so for alcohol consumption; the authors have correctly reasoned this finding to the low power of the alcohol consumption analysis, as only 0.3% of the variation in alcohol consumption is explained by the known GWAS identified genetic variants. The analysis is straightforward and the paper is well-written. I have only a few minor comments:

1. In the Introduction, second paragraph, where the authors describe the alcohol and cancer observational literature, they should add that the evidence is also strong for high alcohol consumption and risk of stomach and colorectal cancer.

2. The authors performed an analysis for lifetime smoking using a UK Biobank GWAS to define relevant instruments. They have already commented in the Discussion about potential bias caused due to overlap of the exposure and outcome datasets. It would be nice to extend the discussion to situations where the overlap is complete.

3. Cancer is a heterogeneous set of diseases, and it doesn't make a lot of sense to conduct analyses for all cancer sites combined.

4. The contamination mixture method developed by one of the study authors could be another method that the investigators could use for smoking initiation to investigate potential pleiotropy but also whether the 361 SNP IV could be subgrouped to more than one potential mechanisms. 

Reviewer #3: This study uses the approach of Mendelian randomization (MR) to assess causality for associations of cancer with smoking and alcohol consumption, respectively. While this is an interesting and worthwhile study and the methodology seems to be sound, I feel that the authors somewhat overestimate the explanatory power of their study. Given that causality has already been established for many of the associations based on comprehensive reviews of experimental and observational evidence by renowned institutions (International Agency for Research on Cancer, US Surgeon General, World Cancer Research Fund…), it seems weird when the authors question causality of associations to justify their MR approach, even though they can only provide low precision estimates due to the small amount of variance explained by their genetic instruments despite high sample sizes and thus cannot exclude false-negative findingns. Having said that, their approach is indeed worthwhile and could be used to further underpin available evidence. But I nevertheless would recommend authors to tone down some of their statements. For example on page 9 of the discussion, where authors state that their findings "extend the observational evidence that smoking is also a risk factor for cancers of the head and neck, oesophagus, stomach, pancreas, cervix, bladder, and kidney." The causality of these associations has been already well-established for years and decades. The authors go on to say that their results offer "strong support for smoking to be considered as a risk factor for this wider range of cancers in clinical practice". I think we are already way past this point - the long available evidence should have already been translated into clinical practice years ago.

The study in its current form is missing one important piece of the puzzle: there is no data supporting whether in this specific sample the genetic instruments are actually associated with higher smoking and alcohol consumption, respectively. This would be further confirmation for the suitability of the genetic instruments, but it would also allow for better assessment of the statistical power to detect effects. Some information on the amount of variance explained by the genetic instruments is reported in the manuscript, but my understanding is that those stem from published meta-analyses. I would thus recommend to add some data on the associations between the genetic instruments and different indicators of smoking and alcohol exposure in the sample.

[LINK]

---

## [Decision Letter · Decision Letter 1]

18 May 2020

Dear Dr. Larsson,

Thank you very much for re-submitting your manuscript "Smoking, alcohol consumption and cancer: A Mendelian randomisation study" (PMEDICINE-D-19-03758R1) for review by PLOS Medicine.

I have discussed the paper with my colleagues and the academic editor and the revised version was seen by two reviewers. There are remaining editorial and production issues need to be dealt with before we would be able to accept the paper for publication in the journal. In particular, we require that you please temper instances of strong causal language throughout the manuscript, provide sufficient data access information in the data availability statement, and include participant summary demographic information.

[LINK]

If you have any questions in the meantime, please contact me (cmoyer@plos.org) or the journal staff on plosmedicine@plos.org. 

We look forward to receiving the revised manuscript by May 25 2020 11:59PM. 

Sincerely,

Caitlin Moyer, Ph.D.

Associate Editor 

PLOS Medicine

plosmedicine.org

Requests from Editors:

1.Response to Reviewer comments: As requested by the reviewer, please include your caveat regarding Reviewer 1, Point #4, a a discussion point in the manuscript (perhaps in the section describing limitations).

2. Title: Please mention the study population in the title (e.g. the UK Biobank, consortia). Please revise your title, we suggest: “Smoking, alcohol consumption and risk of cancer: A Mendelian randomisation study in UK Biobank and international genetic consortia participants” or similar. 

3. Competing Interests: Please add this statement to the manuscript's Competing Interests in the manuscript submission form: "SB is a paid statistical consultant on PLOS Medicine's statistical board."

4. Data Availability: Thank you for providing the summary-level data as a supporting information file. However, we require that you please make the de-identified primary data available, or provide contact information for interested researchers to apply for access to such data (please note that the contact for data access cannot be one of the study’s authors).

PLOS defines the “minimal data set” to consist of the data set used to reach the conclusions drawn in the manuscript with related metadata and methods, and any additional data required to replicate the reported study findings in their entirety. Authors do not need to submit their entire data set, or the raw data collected during an investigation. Please submit the following data:

The values behind the means, standard deviations and other measures reported;

The values used to build graphs;

The points extracted from images for analysis.

5. Prospective analysis plan: Did your study have a prospective protocol or analysis plan? Please state this (either way) early in the Methods section.

6. Abstract (and throughout manuscript): Early in the abstract, the wording could be adjusted to acknowledge that there is, in fact, little or no doubt that smoking causes lung cancer. Also, and more generally throughout the paper, the wording should accommodate existing knowledge more realistically (e.g., the ACS website states quite unambiguously that "Smoking is the most important risk factor for bladder cancer. Smokers are at least 3 times as likely to get bladder cancer as non-smokers. Smoking causes about half of all bladder cancers in both men and women.") Perhaps "genetic predisposition to smoking" is an imperfect proxy for actual smoking. 

7. Abstract: Introduction: Please revise the final sentence to: “Mendelian randomisation study sought to investigate associations in support of a causal relationship between smoking and alcohol consumption and 19 site-specific cancers.”

8. Abstract: Methods and Findings: Please identify some demographics of the participants included in the study (country, etc.) for the genome-wide association studies consortia.

9. Abstract: Methods and Findings: Please clarify that the associations between smoking and prostate cancer in the UK Biobank, and between overall cancer and alcohol consumption did not reach statistical significance. Please provide the confidence intervals and p-values for: “A positive association between alcohol consumption and lung cancer was observed in the International Lung Cancer Consortium, but not in UK Biobank.”

10. Abstract: Methods and Findings: Please revise this sentence to reflect that there was no statistically significant relationship between alcohol consumption and overall cancer, i.e. “no evidence” rather than “limited evidence”: “We found limited evidence that genetically-predicted alcohol consumption was associated with overall cancer (n=75 037 cases; OR 0.95; 95% CI 0.84-1.07; p=0.376).”

11. Abstract: Methods and Findings: In the last sentence of the Abstract Methods and Findings section, please describe the main limitation(s) of the study's methodology.

12. Abstract: Conclusions: Please revise the first sentence to: “Our findings support the well-established relationship between smoking and lung cancer... and suggest that smoking may also be a risk factor for cancer of the head and neck, oesophagus, stomach, cervix and bladder.” Please revise the final sentence to: “We found no evidence supporting a relationship between alcohol consumption and overall or site-specific cancer risk.” or similar, to clarify the meaning of “an association...cannot be precluded.” (please also revise this in the final “Conclusions” paragraph of the Discussion section).

13. Author Summary: Under “What did the researchers do and find?”: Please remove “strong or suggestive” from the first bullet point.

14. Author Summary: Under “What did the researchers do and find?”: Please remove “though

several estimates were in the same direction as those reported by observational studies.” as the directions aren’t specifically mentioned, this is not helpful information.

15. Author Summary: Under “What do these findings mean?”: This text should be distinct from the scientific abstract. (Please see our author guidelines for more information: https://journals.plos.org/plosmedicine/s/revising-your-manuscript#loc-author-summary) 

Please revise the bullet points; we suggest; In this study, we observed a relationship between smoking and lung cancer, as well as evidence that smoking may also be a risk factor for cancer of the head and neck, oesophagus, stomach, cervix, and bladder.” For the second bullet point, we suggest: “We found no evidence supporting a relationship between alcohol consumption and overall or site-specific cancer risk.” or similar, to clarify the meaning of “an association...cannot be precluded.” 

16. Methods: “All participants provided informed consent.” Please specify whether consent was written or oral.

17. Methods: Please provide summary demographic information (population and setting, years of inclusion) for the study participants, in the UK Biobank and GWAS consortia.

18. Methods: Please add the following statement, or similar, to the Methods: "This study is reported as per the Strengthening the Reporting of Observational Studies in Epidemiology (STROBE) guideline (S1 Checklist)."

19. Results: Paragraph 2 (and throughout): Please revise “strong or suggestive evidence” and “suggestive evidence” to be more clear- if statistical significance is intended, please indicate that. If clinical significance is intended, please remove the term because there is no clinical aspect in the study and so clinical significance cannot be addressed.

20. Results: Top of page 10: Please remove the word “slightly” from the sentence: “Adjustment for genetic predisposition to smoking using multivariable Mendelian randomization slightly attenuated the association of genetically-predicted alcohol consumption with lung cancer (OR 1.75; 95% CI 1.23-2.49; p=0.002) (S5 Table).”

21. Results: Where you describe relationships between alcohol consumption and site-specific cancers, please make it clear in the text whether these relationships were statistically significant or not.

22. Discussion: First sentence: Please avoid assertions of primacy ("We are the first....") We suggest you temper this with the phrase “To our knowledge…” or similar.

23. Discussion: First paragraph (bottom of page 10): Please revise this sentence, to temper the causal implications: “... offering strong support of causation to previous observational studies.” (e.g., "supporting a causal relationship as consistent with previous observational ..." might be helpful). 

24. Discussion: Please present and organize the Discussion as follows: a short, clear summary of the article's findings; what the study adds to existing research and where and why the results may differ from previous research; strengths and limitations of the study; implications and next steps for research, clinical practice, and/or public policy; one-paragraph conclusion. Specifically, your discussion is missing a final paragraph reflecting on the study’s implications.

25. Conclusion: Please revise this sentence to: “The results of this study support the well-established relationship between smoking and lung cancer, and suggest that smoking may also be a risk factor for cancer of the head and neck, oesophagus, stomach, cervix and bladder. We found no evidence in support of a relationship between alcohol consumption and overall cancer risk, but associations between alcohol consumption and risk of site-specific cancer should be further investigated.” or similar. Note that this paragraph appears to be identical to the abstract conclusions and the Author Summary.

26. Figure 1: In the legend, the “2” in I2 should be superscript.

Comments from Reviewers:

Reviewer #1: Larsson et al. have satisfactorily addressed the majority of my comments.

The only comment that I don't think was fully addressed was Major Comment 4. I think the caveat that Larsson et al. responded with should be added to the manuscript (i.e. that Wald-type methods are unbiased under the null, and that the estimators do not reflect meaningful causal parameters).

Alex Cornish (ICR, London). 

Reviewer #2: The authors have adequately addressed my comments.

Kostas Tsilidis, Imperial College London and University of Ioannina

[LINK]

---

## [Editor Report · Decision Letter 2]

25 Jun 2020

Dear Dr. Larsson, 

On behalf of my colleagues and the academic editor, Dr. Konstantinos K Tsilidis, I am delighted to inform you that your manuscript entitled "Smoking, alcohol consumption and cancer: A Mendelian randomisation study in UK Biobank and international genetic consortia participants" (PMEDICINE-D-19-03758R2) has been accepted for publication in PLOS Medicine. 

PRODUCTION PROCESS

PRESS

PROFILE INFORMATION

Thank you again for submitting the manuscript to PLOS Medicine. We look forward to publishing it. 

Best wishes, 

Caitlin Moyer, Ph.D.

Associate Editor 

PLOS Medicine

plosmedicine.org